# Synthesis of Wrinkle-Free Metallic Thin Films in Polymer by Interfacial Instability Suppression with Nanoparticles

**DOI:** 10.3390/nano13061044

**Published:** 2023-03-14

**Authors:** Maryam Jalali-Mousavi, Samuel Kok Suen Cheng, Jian Sheng

**Affiliations:** College of Engineering, Texas A&M University–Corpus Christi, Corpus Christi, TX 78412, USA; maryam.jalali-mousavi@tamucc.edu (M.J.-M.); koksuen.cheng@tamucc.edu (S.K.S.C.)

**Keywords:** wrinkle-free thin film in polymer, interfacial instability suppression, nanoparticles, scalable microfabrication

## Abstract

Synthesis of a smooth conductive film over an elastomer is vital to the development of flexible optics and wearable electronics, but applications are hindered by wrinkles and cracks in the film. To date, a large-scale wrinkle-free film in an elastomer has yet to be achieved. We present a robust method to fabricate wrinkle-free, stress-free, and optically smooth thin film in elastomer. Targeting underlying mechanisms, we applied nanoparticles between the film and elastomer to jam the interface and subsequently suppress interfacial instabilities to prevent the formation of wrinkles. Using polydimethylsiloxane (PDMS) and parylene-C as a model system, we have synthesized large-scale (>10 cm) wrinkle-free Al film over/in PDMS and demonstrated the principle of interface jamming by nanoparticles. We varied the jammer layer thickness to show that, as the layer exceeds a critical thickness (e.g., 150 nm), wrinkles are successfully suppressed. Nano-indentation experiments revealed that the interface becomes more elastic and less viscoelastic with respect to the jammer thickness, which further supports our assertion of the wrinkle suppression mechanism. Since the film was embedded in a polymer matrix, the resultant film was highly deformable, elastic, and optically smooth with applications for deformable optical sensors and actuators.

## 1. Introduction

There has been a growing interest in fabricating a deformable layered metal–polymer composite (e.g., conductive thin films over compliant substrates) that enables the development of flexible electronics (e.g., stretchable transistors [1], body-conforming strain/stress sensors [2], flexible solar cells [3], and rollable displays [4]), and biomimetic artificial sensors/tissues (e.g., integrated wearable biosensors [5], artificial skins [6] and muscles [7]). The key technology is to synthesize a conductive thin film over a compliant elastomer. Deposition of rigid films on soft substrates often results in their buckling into sinusoidal undulations [8] or wrinkles [9]. First theoretically articulated by Biot [10] and later experimentally observed by Bowden et al. [11], a semi-infinite incompressible neo-Hookian solid sheet will inherently become unstable to a surface-buckling mode at a critical compressive strain (e.g., 33%, [10,12]), which is not material specific [13]. This wrinkling/buckling phenomenon is ubiquitous to many micro-/nano-scale systems with interfaces of materials having different elastic moduli (e.g., acrylamide–methylenebis(acrylamide) [14], polystyrene–polydimethylsiloxane (PDMS) [15], single-crystal silicon over PDMS [16], etc.) and/or those under compressive stresses (e.g., thermal contraction [17], deswelling [1,18]). Although difficult to completely suppress the wrinkling, management methods have been proposed [11,19] by applying an array of well-defined stress-confining boundaries (e.g., micro-scale geometries [11]) or interfacial-stress-altering chemical micro-patches [20]. Later, Rogers and co-workers [16] placed this class of controlling methods over a theoretical framework using systems of single-crystal semiconductor and poly-crystalline metal films covalently bonded to elastomers. The wrinkles’ characteristics (i.e., amplitude and wavelength) can be predicted and controlled via interfacial loading [16,21]. The development of manipulating wrinkle morphologies in film–substrate systems has enabled the fabrication of stretchable and flexible electronics [1,11,15,22,23,24,25]. By using stress relief mechanisms through the creation of inhomogeneous local stress/strain discontinuities [1,15,18,19,26,27], one can tune the patterns and distributions of these wrinkles, but synthesis of a wrinkle-free interface has not been successful. In tandem, prior efforts on fabricating a wrinkle-free film have focused on principles of reducing interfacial stresses or minimizing its strain. Included are: (i) thin film synthesis over a pre-stretched substrate that limits film deformation [16,27]; (ii) transfer printing methods where a film is first synthesized over a rigid substrate and later transferred to an elastomer substrate, which is, again, to limit film deformation during fabrication and transferring [28,29,30]; and (iii) incorporating nanostructures into the film to suppress the wrinkle formation (e.g., nano-mesh [31,32]). Although the concept of manipulating film mobility with nanostructures is intriguing, fabrication is complex and costly. While transfer printing techniques do offer a good synthesis platform, they rely on subtle disparities in adhesion affinities among materials that often result in an incomplete transfer. Synthesizing films over pre-stretched substrates does provide a robust means to eliminate wrinkles over a large area, but elevated residual stresses severely degrade materials’ mechanical response to additional strain during applications.

Here, we demonstrate a robust and scalable layer-by-layer fabrication method to synthesize wrinkle-free metallic thin films in polymers (WiMTiP). This technique, which utilizes nano-scale jammers deposited at the film–elastomer interface to suppress the development of interfacial instability, enables us to synthesize a conductive nano-/micro-meter thin film over a compliant elastomer substrate without any wrinkles and further encases the film with an additional top polymer layer, i.e., a layered polymer–film–polymer composite. Owing to the interfacial jamming, the synthesized film is optically smooth, whilst its embedment in polymer prevents the film from buckling, delamination and fracturing under large strain (e.g., >60%). These traits make WiMTiP composite highly suitable for a wide range of applications, especially in optomechanical devices such as flexible mirrors, optical switches, and tunable opto-strain micro-sensors, etc. The method, based on the interfacial jamming principle but not on specific material properties, is readily applied to a wide variety of materials (e.g., metals and elastomers in any pairing configurations). Since the synthesis only involves layer-by-layer techniques, a large multi-layer composite (e.g., PDMS-Al-PDMS-Cu-PDMS) with complex patterns containing a wide range of scales (e.g., nm to m) can be easily fabricated. This technique provides a robust platform to study a layered metamaterial composite and inspires new interests in interfacial phenomena at nano metamaterials.

We will first discuss the fabrication principle, followed by a demonstration of its effectiveness via a collection of assorted samples and devices. Using nm aluminum films over PDMS substrates coated with parylene-C jammers as a model, we provide quantitative evidence to further elucidate the underlying mechanism by directly quantifying jamming effects and measuring the interfacial viscoelasticity. Measurements reveal that nm- thick jammers increase the elastomer’s interfacial elasticity by orders of magnitude but maintain its viscoelastic properties. Details on synthesis and characterization procedures are provided in the Appendix A.

## 2. Materials and Methods

### 2.1. Layer-by-Layer Synthesis of a Single-Layer WiMTiP Composite 

We selected PDMS (Sylgard 184, Corning Inc., New York, NY, USA, $E_{PDMS}=300\;kPa-3\;MPa$) as substrate (bottom) and encasing (top) polymer, parylene-C ($E_{PR}=2.76\;GPa$), as the jammer, and various metals (e.g., Al, Ti, Cu, etc.). The composite, consisting of thick PDMS substrate, nm jammer, nm metallic film, and μm PDMS (from bottom to top) was synthesized using a layer-by-layer procedure over a microscope slide (75 × 25 × 1 mm^3^). The following describes the procedure in a sequential order. First, a commercially available microscope slide used as a substrate was cleaned in a Piranha solution (H_2_SO_4_(%):H_2_O_2_(%) = 3:1) for 30 min, then rinsed in deionized (DI) water and air-dried with nitrogen. The slide was backed on a hot plate at 150 °C for 5 min to remove the water molecules completely. PDMS elastomer was mixed at the monomer to crosslinker ratio of 10:1 (*w/w*) and degassed in a desiccator. The elastomer mixture was then poured onto the slide to form the first layer of WiMTiP. The volume was controlled to allow the mixture to pin at the slide’s edges. It is found that pinning-induced menisci reduce residual stresses at sharp edges, which often cause fractures in a rigid film later deposited over it. After the first layer was cured for at least 48 h at 62 °C, jammer (parylene-C) at various thicknesses was deposited onto PDMS using physical vapor deposition (Parylene 2010 Labcoater, Specialty Coating Systems Inc., Indianapolis, IN, USA) at room temperature under vacuum. Although a parylene thickness greater than 0.5 um can be controlled by adjusting the weight of dimers, depositing a parylene layer thinner than 500 nm can be achieved by controlling the vapor concentration during deposition. To achieve this, PDMS substrate was placed in a secondary sealed chamber with a single diffusion orifice, through which the parylene concentration in the secondary chamber could be controlled through the orifice diameter, primary chamber concentration, and deposition time. Using orifices of 0.5, 1, and 1.5 mm in diameter, parylene thicknesses of 7–500 nm can be accomplished. After the jammer was deposited, a 50 nm metal (e.g., Al, Ti, Cu, etc.) film was deposited using an AJA sputtering system (ATC 2000, AJA International Inc., North Scituate, MA, USA) over the jammer layer at 250 W for 6 min. Finally, a μm thick PDMS layer is spincoated over the metal film to encase it. It is worth noting that the nm parylene-C jammer layer deposited via physical vapor deposition at lower temperatures and smaller dimer concentrations (e.g., current deposition conditions) was not a continuous smooth film, which contains many pinholes (Appendix A) and submicron island-like aggregates (Appendix A, [33]). Due to the soft substrate porosity, caulking structures (e.g., structures fingering into the substrate with an average depth of 1.5 μm) have been reported in the literature [33]. These aggregates and caulking structures were confined at the interface within a range of a few microns and made the jammer layer nonhomogeneous. Auxiliary characterization of parylene-C “film” (Appendix A) in comparison with the underlying glass substrate (Appendix A) confirmed these micron- and nano-scale particle-like structures render inhomogeneous interface topology and stress distribution, analogous to that of nanoparticles trapped interfaces (e.g., [34,35], etc.)

### 2.2. Structures of WiMTiP Composites 

Those composites used in Figure 1 and Figure 2 consisted of a 1.5 mm PDMS bottom layer, a 0.5 μm jammer layer (parylene-C), a 50 nm metal film as the middle layer, and a 25 μm PDMS top layer. Samples used in the study on characteristics of surface wrinkles (Figure 3) were constructed with only the bottom three layers, e.g., a 1.5 mm PDMS layer, a jammer layer with various thicknesses (t=0,40,74,118,160,230,&340 nm), and a 50 nm Al film, while samples used in interfacial rheology study (Figure 4) consisted of two bottom layers with and without Al films.

### 2.3. Macroscopic Surface Topology by Optical Microscope 

Wrinkle characteristics (wavelength, λ, and roughness, $k^{*}$) were measured directly using microscopy and atomic force microscopy. Macroscopic roughness patterns (Figure 3A–C) were first visualized by an optical metallurgy microscope (Nikon VT200, Nikon Inc., Tokyo, Japan) at a magnification of 20×. Three randomly selected locations per sample were imaged. The microscopic characteristics (e.g., λ and $k^{*}$) of the same samples were further scanned by an atomic force microscope (AFM Workshop TT-2, AFM Workshop, Hilton Head Island, SC, USA). Three randomly selected areas of 50 μm×50 μm per sample were scanned with a probe (ACLA-10, AFM Workshop, Hilton Head Island, SC, USA) of a 6 nm tip. Scans were visualized and data were extracted by Gwyddion v2.62 (www.gwyddion.net, accessed on 23 January 2023).

To determine the spatial characteristics of the surface roughness, both micrographs and AFM scans were transformed into the corresponding 2D spectra using a non-windowed DFT. The 2D spectra was then be mapped into the polar coordinates, from which the radial power spectral density (rPSD) distributions (Figure 3G) were computed by averaging along the azimuthal direction. The peaks identified in the rPSDs correspond to characteristics mean wavelength of surface wrinkles. Note larger wavelength corresponds to peaks closer to zero. The surface roughness, $k^{*}$, of samples was approximated using AFM scans. The scans were first corrected by fitting a 2nd-order polynomial surface to approximate the base. The $k^{*}$s were calculated as the standard deviation.

### 2.4. Microscopic Surface Characterization by Atomic Force Microscopy (AFM) 

Microscale surface roughness was characterized using AFM (TT-2 AFMWorkshop, AFM Workshop, Hilton Head Island, SC, USA) in tapping mode, while the macroscale roughness was analyzed using spectral analysis of low-magnification micrographs (Figure 3). After AFM surface scans were obtained, the surface profiles were transformed into spectra using non-windowed FFT. The normalized spectra of WiMTiP samples with various jammer thicknesses, t, are shown in Appendix A. The spectra of the wrinkled samples with t<150 nm contains secondary spectra peaks, which shows the characteristic spacing of the wrinkles. Note that those peaks are located within an annular ring showing that the orientation of the wrinkles is isotopic (i.e., wrinkles are oriented in all directions but with single-wrinkle spacing). At t>150 (Appendix A), the spectra show only a single peak corresponding to the DC component (i.e., a smooth surface).

### 2.5. Interfacial Rheology by Nano-Indentation 

The material property of PDMS interfaces “doped” with various jammer thicknesses was measured using AFM based nano-indentation experiments. Measurements were made using (TT-2, AFMWorkshop, AFM Workshop, Hilton Head Island, SC, USA) with an aluminum-coated contact probe. 

To elucidate the effects of jammers on the interfacial rheology, we perform nano-indentation experiments on an elastomer layer (e.g., 1.5 mm thick PDMS over a glass substrate) with a NP jammer layer of various thicknesses (e.g., 0, 40, 130, 340 nm). Since we only wish to emphasize the rheological effects of a jammer layer on the interface (e.g., interfacial viscoelasticity), the two additional top layers in a WiMTiP composite, i.e., nanometer metallic thin film and the encasing PDMS top layer, are conveniently absent. Appendix A shows the sample force–deformation (F−δ) curve with the maximum deformation of 1500 nm and 500 nm for a PDMS layer without and with a jammer layer respectively. The probe indented the specimen at the speed of 500 nm/s. The spring constant of the probe was measured by Sader’s method [36,37] Results show that as the jammer thickness increased, the interface became stiffer and had less hysteresis (or less viscoelasticity). Table 2 summarizes interfacial elasticity approximated by fitting the hertz model over the extension curve. Note that with merely a 340 nm thick jammer, the elasticity, $E_{0}$, increased by two orders of magnitude (e.g., >200 times), which supports our assertion that NPs suppress interfacial instability by substantially increasing the stiffness of the interface. Note further that these effects were confined at the interfaces with little impact on the bulk rheology of the WiMTiP composite.

## 3. Results

### 3.1. Principle of Fabricating Wrinkle Free Metallic Thin Film in Polymer (WiMTiP) by Interfacial Jamming

During deposition (e.g., sputtering, thermal evaporation) of metallic thin film (yellow in Figure 1A), as energetic particles were adsorbed or condensed onto the surface [38], elastomer (blue in Figure 1A) rapidly heated up and expanded. After deposition, the polymeric substrate contracted as it cooled down; meanwhile, the film experienced a compressive strain larger than a critical value and consequently buckled to form wrinkles. Figure 1A (Left) graphically illustrates such a wrinkle formation process in a film-polymer bilayer system, which resulted in complex wrinkle patterns (e.g., a 50 nm amorphous Al film sputtered over a PDMS substrate, as shown in Figure 1B,D).

Existing techniques based on wrinkle management principles have been achieved with various degrees of success. Common strategies include: (i) preloading the elastomer substrate, which introduces a deformation bias to effectively limit thermal strain below a critical value. These methods are effective but leave substantial residual stresses at the interface that limit its range of mechanical responses to large deformation, i.e., under large strain, the film often fractures and delaminates; (ii) minimizing the stress by transferring a pre-synthesized film on a rigid substrate to an elastomer. Although simple in principle, the transferred film often contains defects. For instance, Figure 1C shows that a 50 nm Al film synthesized initially over a dry resist- (Riston^®^, DuPont, Wilmington, DE, USA) covered Si wafer and later transferred to PDMS possesses many fractures; and (iii) prompting covalent bonding between film and substrate, such as plasma activation before deposition. We have found the positive effects are often short-lived (e.g., ~10 min). The failure of the abovementioned methods and our need to develop an optomechanical nano-strain sensor has inspired the current work. Careful examination of our failed attempts has led us to conclude that wrinkling is caused by interfacial instability where past methods have failed to address the underlying mechanism. As pointed out by Herbert Kroemer: “Often, it may be said that the interface is the device” [39,40], we devised a simple method to apply nano-jammer particles to directly suppress these instabilities and consequently allow a simple layer-by-layer synthesis of wrinkle-free thin films in polymer. Elucidated graphically in Figure 1A (right) and demonstrated in Subramaniam et al. [35], the steric jamming of interfacially trapped particles or aggregates confers the solid-like behavior (i.e., increased stiffness) of the jammed interfaces. Subramaniam et al. also pointed out that inhomogeneous states of interfaces produce plastic response [35], and consequently confine the shear deformation locally near particles/aggregates. This steric jamming phenomenon by nanoparticles has been confirmed at polymer–polymer interfaces [41,42]. We chose parylene-C as the jammer, since the parylene-C jammer layer has superior optical clarity but contains inhomogeneously trapped nanostructures that render plastic response, and it can be produced cheaply in scales. Additionally, owing to their nm thickness, the jammers affect only the interfacial rheology but have no distinguishable impact on bulk material properties.

### 3.2. Layer-By-Layer WiMTiP Synthesis

Elucidated graphically in Figure 1A (right), after casting and curing a polymer layer we deposited a layer of jammer particles (e.g., parylene-C using physical vapor deposition, PVD), before a metallic thin film (e.g., *Al, Cu, Ti*) was deposited either amorphously (e.g., sputtering) or by using a semi-crystalline (e.g., thermal evaporation) method. These nanoparticles will “jam” the interface such that interfacial deformation is confined locally and below the film’s critical strain, and consequently prevent it from wrinkling. An additional thin layer of polymer was casted over the film to provide protection and prevent delamination under large strains. A sample of 50 nm thick wrinkle-free Al thin film was sandwiched between two PDMS layers (Figure 1E). From top to bottom, the composite was composed of a 25 μm PDMS, 50 nm Al, 500 nm parylene-C, and 1.5 mm PDMS. Note the jammer thickness was only an estimate obtained by measuring the layer concurrently deposited over a Si wafer co-located with the sample. Superior optical characteristics are evident in Figure 1F, where the sample is held against the cleanroom background and a sharp reflection of a cell phone is observed. Hereinafter, we denote a WiMTiP composite as: *polymer*-*metal*-WiMTiP with abbreviation of the encapsulating polymer followed by the symbol of the metal. If different metal films are synthesized into the same polymer, only the polymer will be provided (e.g., *pdms-*WiMTiP).

Assorted WiMTiP composites and devices have been fabricated (Figure 2) to demonstrate the method’s robustness. Shown in Figure 2A–C, once the elastomer surface was stabilized by jammers (e.g., parylene-C), a wide range of rigid thin films (e.g., Al, Ti, Cr, and Cu in Figure 2A–D, respectively) with different patterns could be easily synthesized over various substrates (e.g., PDMS on glass in Figure 2A, and PDMS wells on a Kapton sheet in Figure 2B) using established deposition techniques (e.g., sputtering with a shadow-mask in Figure 2C). Note that all composites (Figure 2A–D), regardless of materials and deposition methods, were wrinkle-free and possessed mirror-like optical smoothness (e.g., Figure 1F). Being a simple layer-by-layer technique, it is capable of synthesizing multi-layer WiMTiP. As demonstrated in Figure 2D, a two-layer WiMTiP (e.g., PDMS-Cu-PDMS-Cu-PDMS) was fabricated. Note the bottom WiMTiP was a uniform film, while the top was a Cu film consisting of a 2-mm polka dot array. This result highlights that the method is not material specific and applicable to existing deposition techniques (e.g., sputtering, thermal evaporation, vapor deposition, etc.). WiMTiP technology has been used as an opto-mechanical sensor material and already been integrated with multiple microfluidics (Figure 2E–G). For instance, a *pdms-Al-*WiMTiP was integrated as the bottom wall of a microfluidic device (Figure 2E) to detect circulating tumor cells in a patient’s whole blood. Figure 2F shows a WiMTiP device used to investigate the conductivity of a flexible wire, whilst an ecology-on-a-chip (*eChip* [43,44]) with WiMTiP sensor (Figure 2G) was used to study bacterial biofilm adhesion.

### 3.3. Effect of Jammer

To understand the underlying wrinkle-prevention mechanism by interfacial jamming, we examined the effects of the jammer thickness on the morphology and rheology of a *pdms*-*Al*-WiMTiP. We systematically varied the thickness of the jammer at the interface and characterized the surface roughness (e.g., wavelength and amplitude) by optical microscopy and AFM. Rheology of these “jammed” interfaces (e.g., Young’s modulus and viscoelasticity) was examined by nano-indentation.

To elucidate the proposed suppression mechanism by interfacial jamming and to determine the critical jammer thickness, we first deposited a 50 nm aluminum film over a 1 mm PDMS pre-coated with various thicknesses (e.g., t=0−1500 nm, or t=0,40,74,118,160,230,340,500,1000, and 1500 nm) of jammers (e.g., parylene-C in this study). It is worth emphasizing that thickness is a rough estimation. Due to the porous nature of the polymer, jammer particles are expected to penetrate the substate. Here, we only wish to use these purported thicknesses for comparison purposes. In contrast to normal WiMTiP composites, we forgo the top polymer layer over the thin film so that interfaces can be probed properly. Micrographs (Figure 3A–C) of samples with jammer thicknesses (t=0,74,160 nm) reveal that well-organized wrinkles are formed when the jammer layer is thin (Figure 3A,B), while wrinkles are uniformly suppressed when the jammer exceeds a critical thickness (e.g., ~160 nm, Figure 3C). These wrinkles are randomly oriented but formed in a labyrinth-like pattern (Figure 3 and Appendix A). Yu et al. [45] has pointed out that, owing to the equal-biaxial stresses, the interface energy minimization favors omni-directional wrinkles (Appendix A) with a “single” wavelength (λ) (Figure 3H). Using an AFM (AFMWorkshop, HD-2, AFM Workshop, Hilton Head Island, SC, USA), surface profiles of selected WiMTiP samples with t=0,74,118,340 nm revealed that the spacings between two adjacent wrinkles increased monotonically with t, but were completely suppressed after a critical thickness. Note that the wrinkle suppression occurred precipitously after t = 160 nm (Figure 3H), while the topological change of wrinkles was gradual and exhibited a strong dependency on t. Radial power spectrum density (PSD) distributions (Figure 3H) computed from micrographs of t=0,40,74,118,160,230,340 nm(selected ts in Figure 3A–C) elucidate the existence of characteristic wavelengths (peaks in Figure 3H) and their correlations to the jammer thickness. Note that, since PSDs are plotted with respective wavenumbers (i.e., 1/λ), the closer a peak is located to the *y*-axis the larger the wrinkle spacing (λ) is. The characteristics (e.g., λ and surface roughness, $k^{*}$) of wrinkles estimated using both microscopy and AFM is summarized in Table 1 and graphically presented in the inset of Figure 3H. Analysis on λ by AFM and microscopy agreed well with each other. $k^{*}$ ranging from 75 nm (for t=0 nm) to ~144 nm (for t=118 nm) increased with the jammer thickness (t), while once t exceeded a critical thickness (e.g., $t^{*}=$ ~150 nm), $k^{*}$ measured at values <20 nm represented an optically smooth surface. Observations of the proportional dependency of $k^{*}$ on t are in good agreement with observations by [46] using a different bi-layer polymer system (i.e., nm polytetrafluoroethylene/carbon nano-tube film over mm PDMS substrate). The critical thickness, $t^{*}$, would vary strongly with the elastomer and jammer. In the current case, with PDMS and parylene-C, $t^{*}$ was estimated to be ~150 nm.

To elucidate rheological implications of jammers at the interface, we have performed nano-indentation experiments to characterize the interface elasticity. For comparisons, three categories of samples were fabricated. They were PDMS only (red in Figure 4, control), PDMS with jammers of various thicknesses (t=40,130,148,340 nm, blue in Figure 4), and a smooth 50 nm Al film over PDMS with jammers thicker than $t^{*}$ (t=160,340 nm, green in Figure 4). Note that no wrinkles were formed in all samples. F−δ curves revealed that, in comparison to PDMS alone, composites with jammers were significantly stiffer (Table 2, Appendix A). Results (blue in Figure 4) demonstrate two regimes: as $t<t^{*}$, F−δ curves exhibited pronounced telltale shapes of a viscoelastic interface (i.e., parabolic) similar to that of the PDMS sample, while the interface became more elastic and stiffer (i.e., linear), as t exceeded $t^{*}$. Also note that the transition between these two interfacial behaviors was rapid and binary-like (i.e., curves were segregated into two well-defined regimes). This trend persisted in samples with both Al film and jammers. Note that additional Al film increased the interface elasticity slightly, but jammer characteristics remained unchanged. The observations strongly support our assertion that microscopic jammers “jam” the interface to confine the strain locally and consequently prevent instability from developing; macroscopically, they increase the interfacial elasticity to withstand stresses during synthesis.

### 3.4. WiMTiP Sensitivity for Optomechanic Sensing

It was found that, owing to its residual stress-free fabrication, WiMTiP composite was highly sensitive to minute strain. Demonstrated anecdotally in Appendix A, a WiMTiP composite of 1.5 mm–PDMS, 500 nm jammer, 50 nm Al film, and 10 μm agar, in the order from bottom up, has been fabricated. Due to the dehydration of agar after fabrication, complex roughness patterns can be observed on Al film at the beginning of Appendix A. When a sessile water drop was placed on the top of the agar layer, the hydration of the agar relieved residual stress and allowed the recovery of film topology to that of optically smooth film. Since the elastic modulus was only ~1.1 kPa, this anecdotal evidence highlights that WiMTiP is highly sensitive to small stress, and the resultant deformations are reversable.

## 4. Discussion

In summary, we present a scalable method to synthesize wrinkle-free rigid thin film encased in polymer. Contrary to prior works [17,47], the proposed method used nanoparticles (NPs) to jam the interface, and subsequently to prevent wrinkle formation. Using Al film and PDMS as a model system and parylene-C as jammers, we have demonstrated that the deposition of NPs increases the interface elasticity and reduces its viscosity. It is also shown that the “jamming” effect improves progressively when $t<t^{*}$ (e.g., $t^{*}=150\;nm$ for PDMS), after which the instability is suppressed completely and results in optically smooth film in polymer. We also demonstrated above (Figure 2, Appendix A) that the proposed method is simple, reproducible, low-cost, and scalable with potential in nanoscale sensor and actuator applications.

## Figures and Tables

**Figure 1 nanomaterials-13-01044-f001:**
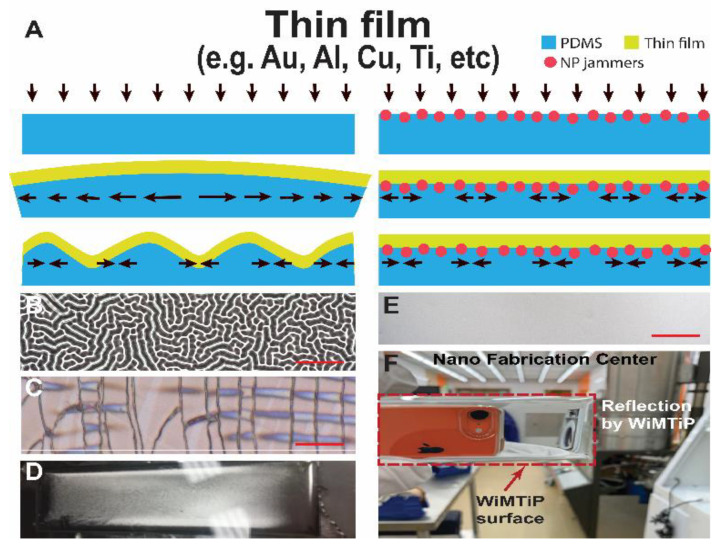
Principle and sample results of fabricating wrinkle-free metallic thin film in polymer (WiMTiP) using nanoparticle (NP) interfacial jamming technology. (**A**) Schematics elucidating: (Left) wrinkle formation mechanism of a thin film over an elastomer substrate by compressive stress-induced interfacial instability. (Right) wrinkle suppression mechanism using NPs to jam the interface and subsequently confine compressive stresses locally. (**B**–**D**) Characteristics of a 50 nm thick Al film over a 2 mm PDMS substrate without jammers. (**B**) Micrograph (10×) of wrinkles developed as the film was deposited onto the substrate. (**C**) Micrograph (10×) of fractures developed as the film pre-deposited over a wafer is transferred to the substrate. (**D**) Image of a glossy (non-reflective) sample. (**E**,**F**) Characteristics of a 50 nm Al in PDMS WiMTiP sample with 500 nm thick jammers. (**E**) Micrograph (10×) of smooth WiMTiP surface. (**F**) Photograph of a WiMTiP sample against cleanroom background showing a reflection of the recording camera. Scale: 20 μm.

**Figure 2 nanomaterials-13-01044-f002:**
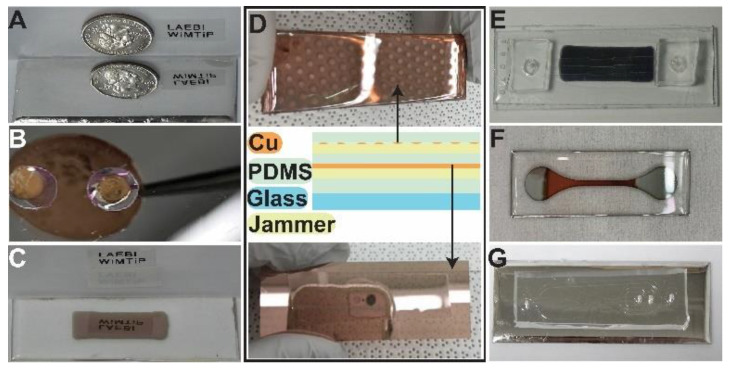
Gallery of various samples and devices using WiMTiP composite: top encasing polymer (25 μm)–middle metal film (50 nm)–bottom polymer substrate (1.5 mm). Jammer (yellow in sketch) is present only between bottom polymer and film. (**A**) PDMS–Al–PDMS on a glass slide, (**B**) PDMS–Ti–PDMS on a polyurethane sheet, (**C**) PDMS–Cr–PDMS nano-strain sensor on a glass slide where rectangular Cr film was sputtered with a shadow mask. (**D**) A double-layer WiMTiP over a glass slide: PDMS (1 mm)–Cu (50 nm: polka dots)–PDMS (25 μm)-Cu (50 nm film)-PDMS (50 μm). Top: encased polka dot Cu film; Middle: Schematics of double-layer WiMTiP; Bottom: encased uniform Cu film. (**E**) A microfluidic channel with a 50 nm *pdms-Al*-WiMTiP for measuring viscoelasticity of cancer cell membrane. (**F**) A *pdms-Ti*-WiMTiP for studying the conductivity of a nm film under large strain. (**G**) *eChip* microfluidics with flow focusing and nano-strain sensor for quantifying adhesion force of a live biofilm in shear flow.

**Figure 3 nanomaterials-13-01044-f003:**
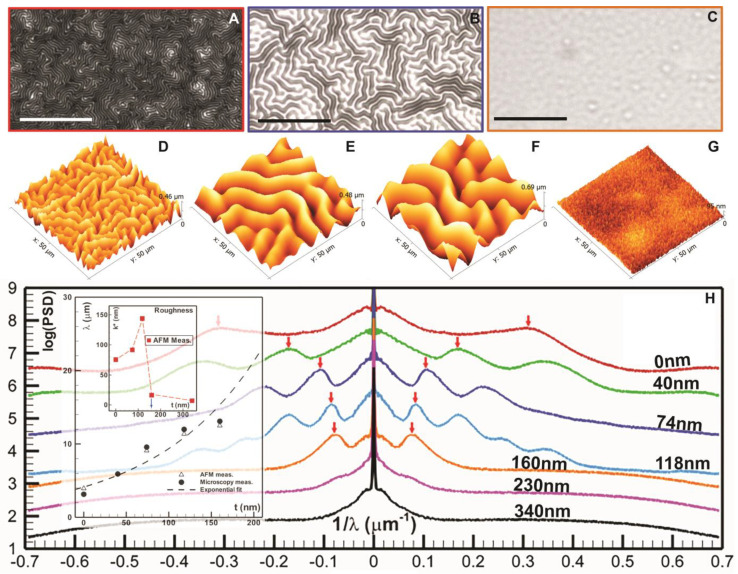
Topological characteristics of a 50 nm Al film over a 1.5 mm PDMS substrate pre-coated with various jammer thickness, t. Top row: micrographs of surface roughness of the Al-PDMS bilayer with t= (**A**) 0, (**B**) 74, & (**C**) 160 nm. Middle row: 50×50 μm AFM roughness images with t= (**D**) 0, (**E**) 74, (**F**) 118 nm, and (**G**) 340 nm. (**H**) radial power spectra density (rPSD) of surface roughness. rPSDs are artificially shifted vertically for clarity. Label: t. Arrows mark the characteristic wrinkle wavelength, λ. Insert: λ vs. t using both micrographs and AFM scans; and characteristic surface roughness $k^{*}$ vs. t using AFM measurements. Scale: 20 μm.

**Figure 4 nanomaterials-13-01044-f004:**
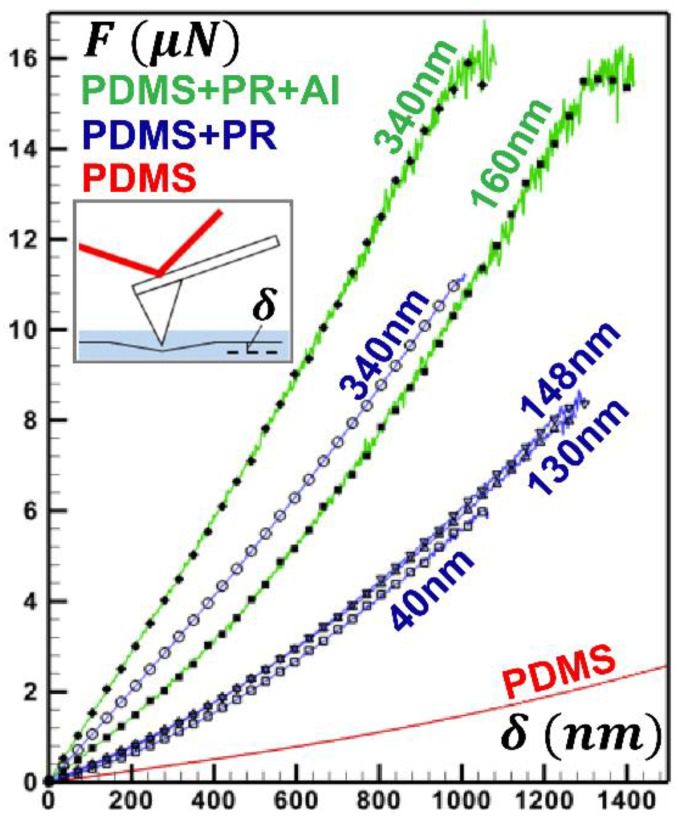
Force (F, μN)-indentation (δ,nm) measurements by AFM on various Al-PDMS bilayers on a glass slide. F−δ curves are color-coded by sample types: Red—1.5 mm PDMS, Blue—a 1.5 mm PDMS substrate with a jammer (parylene-C) of various thickness (t=40,130,148, and 340 nm), Green—a 50 nm Al film over a 1.5 mm PDMS substrate with a jammer of t=160 and 340 nm. Inset: schematics of AFM-based nano-indentation measurement.

**Table 1 nanomaterials-13-01044-t001:** Characteristics of wrinkles in a 40 nm thick Al film over a 1 mm-PDMS substrate coated with various t. Roughness, $k^{*}$, is the first order moment of surface roughness measured by an AFM with a scanning area of 50 μm×50 μm.

Jammer Thickness, t (nm)	$\mathrm{Wrinkle}\;\mathrm{Length},\;\lambda \;\left(\mu m\right)$		$\mathrm{Roughness}\;k^{*}\;\left(nm\right)$
	AFM	Microscope	
0	3.937	3.139	75.460
40	NA	5.904	NA
74	9.091	9.573	91.505
118	11.364	11.991	143.962
160	50	54.162	16.226
230	NA	Inf	NA
340	Inf	Inf	6.947

**Table 2 nanomaterials-13-01044-t002:** Elasticity, $E_{0}$ (MPa), approximated by fit the extension F−δ curve with Hertz linear model, $F\left(\delta \right)=\frac{C_{n}E_{0}}{1-\nu^{2}}\delta^{n}$, where $C_{n}$ is the shape correction of probe tip, n is the power determined by the probe tip shape, and δ is WiMTiP specimen deformation.

Jammer Thickness, t (nm)	$E_{0}\pm \sigma$ (MPa)
0	0.635±0.268
40	17.2±3.836
130	44.911±11.235
340	135.942±23.989

## Data Availability

Data will be available upon valid request.

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
