# Peer review of "Synthesis of Wrinkle-Free Metallic Thin Films in Polymer by Interfacial Instability Suppression with Nanoparticles"

_nanomaterials, 2023, doi:10.3390/nano13061044_

Round 1

Reviewer 1 Report

The manuscript presents an interesting way to prevent buckling of metallic thin films on a compliant substrate such as elastomeric PDMS. The key to this success lies in the use of “interface jamming” by inserting nanoparticles. It has been a notorious phenomenon that stiff film on a compliant substrate undergoes mechanical instability. The authors have systematically investigated the effect of jammer layer thickness and found the transition from viscoelastic to elastic at interface.

Overall, I like this approach, and should be published in Nanomaterials.

Author Response

See attached response.

Reviewer 2 Report

Authors demonstrate a unique method preventing wrinkles from metal/elastomer composite films by inserting a jammer layer of parylene-C. authors investigated the effect of parylene-C on wrinkling (or buckling) of metal thin layer on PDMS substrate by adjusting the thickness of praylene-C. As a result, a large-scale wrinkle-free film in an elastomer was achieved. Even though the effect o of the jammer layer to prevent wrinkling behavior of samples was clear and in good agreement of experimental data, there are several critical issues that should be addressed before its publication in the “nanomaterials”. 

(1) The underlying mechanism preventing wrinkles from the surface in this study is not clear. Parylene-c as a jammer is not a nanoparticle. Authors prepared the continuous layer of parylene-C with various thickness. Even though authors provided a schematic illustration of Figure 1 in order to explain the mechanism, it does not contain enough information for our general readers. I think authors need to improve the schemes and to add more information. If parylene-C as a shape of particle was used in this study, wrinkle-preventing behavior observed in this study might not be dependent on its thickness.

(2) Authors need to compare the bulk mechanical properties of parylene-C and those of PDMS in order to explain how the jamming layer can change (or stabilize) the surface (or interface) properties. Because parylene-C is a kind of polymers, one can think that the surface of PDMS can be stabilized if a continuous polymer layer is prepared instead of the parylene-C. In addition, one think that layers of some polymers which have similar mechanical properties to those of parylene-C also can give same effects. Authors should give a reason why they only used the parylene-C in this study.

(3) I don’t agree that authors express parylene-C as a nanoparticle because of the abovementioned reason. 

(4) To characterize the critical thickness (~160nm) of parylene-C, it seems to me that a plot of wrinkle length (or roughness) versus jammer thickness should be powerful. Otherwise, I think that authors cannot decide the critical thickness of 160nm. 

(5) In Table 2, there should be elasticity (Eo) of jammer thickness of 160nm even though authors claimed it as a critical thickness. 

Author Response

see attached point to point responses

Reviewer 3 Report

The authors report the advances in Synthesis of wrinkle-free metallic thin films in polymer by interfacial instability suppression with nanoparticles in this study. In addition, the proposed using nano-particles to jam the interface and subsequently to prevent wrinkle formation. Using Al film and PDMS for model system and Parylene-C as jammers and NPs deposition increases the interface elasticity and the viscosity reduces were demonstrated. The simple proposed method, reproducible, low-cost, and scalable with potential in nanoscale sensor and actuator applications were also reported. Finally, the authors successfully report the film embedded in a polymer matrix, highly deformable, elastic, and optically smooth for applications in deformable optical sensors and actuators.

My recommendation the manuscript in the present form lacks grammar rigour, however, a minor revision is necessary before considering for publishing in the present form.

Author Response

See attached responses

Round 2

Reviewer 2 Report

All response to the comments raised before were prepared properly. The changes made by authors in this revised manuscript are adequately supporting their experimental results. Therefore, I think the manuscript is ready for the publication in the Nanomaterials as it is.